# Electroseparation of Slaughterhouse By-Product: Antimicrobial Peptide Enrichment by pH Modification

**DOI:** 10.3390/membranes10050090

**Published:** 2020-05-03

**Authors:** Rémi Przybylski, Laurent Bazinet, Loubna Firdaous, Mostafa Kouach, Jean-François Goossens, Pascal Dhulster, Naïma Nedjar-Arroume

**Affiliations:** 1University Lille, INRA, ISA, University Artois, University Littoral Côte d’Opale, EA 7394 – ICV – Charles Viollette Institute, F-59000 Lille, France; remi.przybylski@polytech-lille.fr (R.P.); loubna.firdaous@univ-lille.fr (L.F.); pascal.dhulster@univ-lille.fr (P.D.); 2 Institute of Nutrition and Functional Foods, Department of Food Sciences and Laboratory of Food Processing and Electromembrane Processes (LTAPEM), Laval University, Quebec, QC G1V 0A6, Canada; laurent.bazinet@fsaa.ulaval.ca; 3University Lille, CHU Lille, EA 7365 – GRITA – Groupe de Recherche sur les formes Injectables et les Technologies Associées, PSM (Plateau de Spectrométrie de Masse), F-59000 Lille, France; mostafa.kouach@univ-lille.fr (M.K.); jean-francois.goossens@univ-lille.fr (J.-F.G.)

**Keywords:** electrodialysis, ultrafiltration membrane, antimicrobial peptide, waste valorisation

## Abstract

The fractionation of bioactive peptides from hydrolysate is a main challenge to produce efficient alternative for synthetic additives. In this work, electrodialysis with ultrafiltration membrane (EDUF) was proposed to increase the purity of one antimicrobial peptide from slaughterhouse by-product hydrolysate. This targeted-peptide, α137–141 (653 Da, TSKYR), inhibits a large spectrum of microbial growths and delays meat rancidity; therefore, if concentrated, it could be used as food antimicrobial. In this context, three pH values were investigated during EDUF treatment to increase the α137–141 purity: 4.7, 6.5, and 9. pH 9 showed the highest purity increase—75-fold compared to the initial hydrolysate. Although the whole hydrolysate contains more than 100 peptides, only six peptides were recovered at a significant concentration. In this fraction, the α137–141 peptide represented more than 50% of the recovered total peptide concentration. The EDUF α137–141-enriched fraction obtained in this optimized condition would be a promising natural preservative to substitute synthetic additives used to protect food.

## 1. Introduction

In the last decade, bioactive peptides isolated from complex hydrolysates have gained considerable attention from the scientists and industrials [1,2,3]. Moreover, these peptides extracted from hydrolysate by-products are highly valued molecules with various bioactivities such as antihypertensive, antidiabetic, or antimicrobial effects [4,5]. However, the isolation of such a bioactive peptide is very challenging due to the complex hydrolysate containing various peptide sequences with numerous physicochemical properties. Thereby, some studies have focused on the recovery of bioactive fractions from a complex source [6,7]. In this way, some peptides were enriched in a fraction that had a potential for industrial applications, depending on its described activity. Nevertheless, the peptides contained in a fraction were mixed with other peptides with different bioactivities, and some of them interfered with the targeted bioactivity or decreased it. However, it is known that increasing peptide purity allows for better bioactivity [8]. 

Pressure-driven membrane filtration (such as ultrafiltration) is the most commonly used strategy to fractionate a hydrolysate. The related advantages are their low cost and high productivity compared to the chromatographic techniques [9]. To improve the separation process selectivity, electrodialysis with ultrafiltration membranes (EDUF) was developed and has been recently patented [10]. EDUF separates molecules in accordance with their charge (by application of an electrical field as driving force) and molecular weight (by the cut-off of the filtration membrane). Hence, many peptide fractions demonstrating bioactivites were successfully fractionated by EDUF from different hydrolysate sources such as salmon [11] or flaxseed [12]. Notwithstanding this, to the best of our knowledge, no study has focused on the specific separation behaviour of one peptide from a protein hydrolysate during its separation by EDUF to improve its purity.

In this context, bovine blood represents a potential interesting and abundant slaughterhouse by-product [13]. After its centrifugation, the red fraction contains mainly haemoglobin, broadly described as a rich source of antimicrobial peptides after enzymatic hydrolysis [14]. Among them, there is an interesting peptide representing the five last amino acids of the bovine haemoglobin α-chain. It is the α137–141 peptide (653 Da, TSKYR, pI 10.5), an hydrophilic antimicrobial peptide with a large antibacterial spectrum, specifically against pathogenic bacteria commonly responsible for food alteration [15]. Recently, its application on meat demonstrated its potential as a food preservative. Indeed, the α137–141 peptide could be a natural alternative for synthetic additives by delaying lipid oxidation and inhibiting microbial growths [16,17].

Therefore, this present work proposed a first approach to produce a peptide fraction enriched in α137-141 peptide by using EDUF and by regulating the hydrolysate pH. Indeed, peptide charges are influenced by the hydrolysate pH. Indeed, by studying different pH, peptide charges can change (positively if pH < pI and negatively if pH > pI) and then will influence migration through the ultrafiltration membrane. In this way, the number and concentration of peptides being able to migrate in the recovery compartment will change with the hydrolysate pH. Consequently, the EDUF treatment selectivity should be influenced by the pH control. Moreover, there was a focus on the analysis of the peptide transfer from the feed of the electrodialysis to try to explain their migration.

## 2. Materials and Methods 

### 2.1. Chemicals and Standard

All chemicals and solvents were of analytical grade and from commercial suppliers Sigma-Aldrich (Saint-Quentin Fallavier, France) or Flandres Chimie (Villeneuve d’Ascq, France). The purified bovine haemoglobin (BH) was also supplied by Sigma-Aldrich. The ultrapure water was prepared using a Milli-Q system. The standard α137-141 was provided by Genecust (Ellange, Luxembourg).

### 2.2. Hydrolysis

A stock solution was prepared by adding 15 g of BH into 100 mL of ultrapure water. After centrifugation (30 min at 4000 min^−1^), the supernatant was kept and its concentration in BH was assessed by the Drabkin’s method [18]. A BH concentration of 1% (*w*/*v*) was obtained by dilution of stock solution with water.

The pepsin from porcine gastric mucosa (EC 3.4.23.1, 3200–4500 units·mg^−1^ protein) was the chosen enzyme to digest BH using a ratio enzyme/substrate of 1/11 (mole/mole) under a constant pH of 3.5 and a temperature of 23 °C, according to a previous study [19] and the production of the targeted peptide [16]. The hydrolysis was stopped by addition of sodium hydroxide (5 M) up to a final pH of 9 after 30 min, corresponding to a degree of hydrolysis of 5%, assessed by the ortho-phthaldialdehyde method [20].

After hydrolysis, the hydrolysate was decolorized to eliminate the remaining haem because it was shown in a previous study that the presence of haem has a negative impact on the peptide migration during EDUF [21]. Removing the haem consisted of slowly decreasing the pH at a value of 4.7 with HCl (2 M). Both phases were separated by a 30 min centrifugation (4000 min^−1^), and the supernatant containing the peptides maintained in solution was kept. After that, the hydrolysate was ready for EDUF electroseparation.

### 2.3. Electroseparation

#### 2.3.1. Equipment

The electrodialysis cell was a MicroFlow type cell (Electrocell AB, Täby, Sweden) with an effective membrane area of 10 cm^2^. The cell configuration is displayed in Figure 1. An EDUF cationic configuration was chosen first to recover specifically the positive charged compounds Secondly, in the case of α137–141 purification from hydrolysate, this configuration allowed the removal of the maximum of other peptides, mainly peptides that were negatively charged.

Four closed compartments were delimited by the different membranes. In this configuration, the anionic and cationic membranes were separated the Na_2_SO_4_ (20 g·L^−1^) electrode rinsing solution. At the beginning of the process, on both sides of the ultrafiltration membrane, the cationic peptide recovery and the feed compartments were circulated and each contained a KCl solution (2 g·L^−1^) and a hydrolysate, freshly prepared. Each compartment was connected to a separate external reservoir to recirculate the solution by using centrifugal pumps (Iwaki, Marcoussis, France). The flow rate of each solution was controlled by its own flowmeter (Kobold Instrumentation, Cergy-Pontoise, France). Finally, the anode/cathode voltage difference was supplied by a variable 0-30 V power source, HQ Power PS3003 (Xantrex, Burnaby, BC, Canada).

The ultrafiltration membrane (UF) used was made of polyethersulfone (PES) with a molecular weight cut-off of 10 kDa (HFK-131, Koch, France). The ion-exchange membranes were Neosepta cationic (CMX-SB) and anionic (AMX-SB) food-grade membranes manufactured by Astom (Tokyo, Japan) and provided by Eurodia (Pertuis, France). This kind of membrane was chosen according to previous studies. Indeed, many studies used this membrane made of PES to separate small active peptides from different feeds by EDUF. These results showed that this material is adapted for peptide separation and to avoid or limit membrane fouling [22]. 

#### 2.3.2. Electroseparation Procedure

All the compartments contained 500 mL of solution (Na_2_SO_4_, KCl or hydrolysate). The voltage was maintained constant all over the process at 20 V, corresponding to an electric field of 9 V·cm^−1^. The voltage applied during EDUF can affect the separation result. The Ohm’s law (U = RI) shows that this parameter could affect the separation [22,23]. In our study, the voltage value was determined by preliminary tests. The peptide transfer though the ultrafiltration membrane was less efficient, with a high value of resistance. It was constant for the study the modification of current intensity and allowed its comparison with our previous work [17].

The flowrates were 12 L·h^−1^ for the electrode solution (Na_2_SO_4_) and 18 L·h^−1^ for the recovery compartment (KCl) and the feed compartments (hydrolysate).

Each electroseparation was carried out in triplicate. Membranes in the EDUF cell were all changed for a new tested condition.

To study the effect of pH on the electroseparation, the compromise was to select different pH values that minimize or avoid the migration of peptides, which can co-migrate with the α137–141, and to keep the electrophoretic mobility as high as possible for the α137–141 peptide (pI about 10.5). Moreover, the hydrolysate pH should not exceed 10 to avoid basic hydrolysis. Thus, the pH was maintained constant at a given value of 4.7 (the native pH value for the hydrolysate without haem), 6.5 (close to the neutral pH but with a high electrophoretic mobility for α137–141), and 9 by manual addition of HCl (0.1 M) or NaOH (0.1 M) in feed and recovery compartments of EDUF. 

After each repetition, the EDUF cell was cleaned to remove a potential deposit in the apparatus.

Membrane integrities were verified after each use of EDUF. No significant decrease in membrane performance or fouling after an EDUF run were observed in our conditions for this study. Moreover, the risk of fouling was minimized because the haemoglobin haem was discarded as described previously [21].

### 2.4. Analyses

#### 2.4.1. RP-HPLC Analyses and Quantification Method for Total Peptides and α137–141 Peptide

The liquid chromatographic system consisted of a Waters 600E automated gradient controller pump module, a Waters Wisp 717 automatic sampling device, and a Waters 996 photodiode array detector (Milford, CT, USA). The applied methodology for analysis and peptide quantification, mainly α137–141, was described in a previous paper [16].

Additionally, the total peptide concentration of each sample from BH hydrolysate was assessed by measuring the total area corresponding to the peptides (Atot) using the Millennium software and correlated with the initial BH concentration measured by using the Drabkin’s method [18].

The α137–141 purity (%) was assessed by A_α137-141_/A_tot_.

The enrichment factor in α137–141 was assessed by (A_α137-141_/A_tot_)_KCl solution_/(A_α137-141_/A_tot_)_hydrolysate_.

#### 2.4.2. Mass Spectrometry Analyses

The LC–MS analyses were performed on a UFLC-XR device (Shimadzu, Japan) coupled to a QTRAP 5500 MS/MS hybrid system triple quadrupole/linear ion trap mass spectrometer (AB Sciex, Foster City, CA, USA) equipped with a Turbo VTM ion source. Instrument control and data acquisition were performed using the Analyst 1.5.2 software. The RP-HPLC separation was carried out on the same column used for the RP-HPLC analyses (§ 2.4.1). The elution was performed with the same gradient, previously described, using formic acid (0.1%) instead of trifluoroacetic acid (0.1%) for the RP-HPLC analyses. MS analysis was carried out in positive ionization mode using an ion spray voltage of 5500 V. The nebulizer gas (air) and the curtain gas (nitrogen) flows were set at 30 psi. The Turbo VTM ion source was set at 550 °C with the auxiliary gas flow (air) set at 50 psi.

The MS/MS analyses were completed with the BioAnalyst 1.5.1 and Peaks 7 software.

## 3. Results and Discussion

### 3.1. Effect of pH Controlled at 4.7

The first step was to control the pH at 4.7 (corresponding to the native value of the discoloured hydrolysate) into the feed (hydrolysate solution) and the recovery compartment (KCl solution). These results were compared to those obtained without pH control.

#### 3.1.1. Electrodialytic Parameter Evolution: pH, Conductivity, and Current Intensity

Without pH control, the pH of the recovery compartment increased from 5.5 to 10 after only 30 min of EDUF corresponding to the appearance time of the limiting current density, previously described [22,23]. Overpassing the limiting current density leads to the phenomenon of water-splitting, and in our case, it occurred mainly at the interface of the cation exchange membrane because the feed pH was stable all along the treatment. Hence, a two-step demineralization was observed by using the pH control (Appendix A). The first step was due to the fast demineralization of the recovery compartment by the leaving of K^+^ and Cl^-^ during the first 30 min of treatment (slope of −1.32 mS·cm^−1^·h^−1^, *r*^2^ = 0.99), as previously observed [23]. After that, the pH into the compartments evolved, requiring a regulation to keep it constant. Therefore, adding HCl contributed to the demineralization slowdown after 30 min of the experiment, with a slope of −0.33 mS·cm^−1^·h^−1^ (*r*^2^ = 0.99). Consequently, the evolution of KCl conductivity without pH control was linear throughout the EDUF treatment, with a slope of −0.34 mS·cm^−1^·h^−1^ (*r*^2^ = 0.99). 

For the hydrolysate compartment, the two working conditions presented similar tendencies in terms of conductivity evolution. Without pH control, the hydrolysate conductivity evolved from 7.6 ± 0.2 mS·cm^−1^ to 10.2 ± 0.5 mS·cm^−1^ (corresponding to a mineralization rate of 34%), and by pH control, the conductivity evolved from 7.0 ± 0.1 mS·cm^−1^ to 8.6 ± 0.3 mS·cm^−1^ (corresponding to a mineralization rate of 24%) (data not shown).

As for the current intensity (Appendix A), the first step of decrease was observed by using the pH control during the first 30 min of EDUF treatment with a slope of −0.19 A·h^−1^ (*r*^2^ = 0.99). After that, the second step had a slope of −0.04 A·h^−1^ (*r*^2^ = 0.86) until the end of the experiment. The current intensity without pH control showed a linear shape throughout the EDUF treatment with a slope of −0.06 A·h^−1^ (*r*^2^ = 0.98), confirming the previous conductivity analyses.

#### 3.1.2. Total Peptide Migration Using pH Controlled at 4.7

Figure 2a presents the evolution of recovered total peptides with and without pH control during 4 h of EDUF treatment. As observed, the peptide migration was higher under pH control. Indeed, the recovery yield was 2.8 times higher for the total peptides (from 38.5 ± 5.5 to 108.4 ± 2.8 mg·L^−1^). Moreover, using pH control allowed the peptide migration to be kept in a linear manner during all the EDUF treatments. On the other hand, the peptide migration slowed down after 30 min of the experiment without pH control. Indeed, without pH control, the total peptide concentrations did not evolve significantly from 2 to 4 h of the experiment (from 33.8 ± 3.9 to 38.5 ± 5.5 mg·L^−1^), contrary to the pH control condition (from 66.0 ± 11.7 to 108.4 ± 2.8 mg·L^−1^).

Figure 3 shows the chromatograms for both conditions after 4 h of EDUF treatment, comparing the two migrated peptide populations. With controlled pH, more peptides were able to migrate through the ultrafiltration membrane. To confirm this fact, the analyses at the end of EDUF treatment by mass spectrometry were carried out. Thereby, it was shown that the number of peptides into the recovery compartment increased when pH was controlled and maintained constant. Indeed, only 16 peptides were recovered without controlled pH, and 40 peptides migrated into the recovery compartment with controlled pH. At the beginning of the experiment, the whole feed contained about 101 identified peptides. The identification of the peptide sequences is summarized in Appendix A.

The identification of recovered peptides was coherent with the whole hydrolysate. In fact, only one peptide with a pI less than 4.7 was identified in the recovery compartment—the EAL (α27–29) peptide with a theoretical pI of 3.81. Its small molecular weight (332 Da) and its proximity of pI with the working pH could be responsible for its migration though the ultrafiltration membrane. However, in accordance with the calculated theoretical pI of peptides, some peptides were able to migrate into the recovery compartment but were not recovered. Due to the zipper mechanism involved in the peptide generation [15,19], the peptide population is constituted by intermediate peptides, giving the final peptides at higher hydrolysis degrees. Hence, high (intermediate) and low (final) molecular weight peptides were mixed in this hydrolysate at a degree of hydrolysis of 5%. Moreover, the hydrolysis kinetics proposed preferential steps to lead to the same peptide [24,25]. With these considerations, the haemoglobin chains could be divided into several parts (Figure 4). The haemoglobin α-chain contained four fragments: α1 (V1 to M32), α2 (F33 to N98), α3 (K99 to L106), and α4 (V107 to R141), and the β-chain comprised six fragments: β1 (M1 to L30), β2 (L31 to F40), β3 (F41 to F84), β4 (A85 to L109), β5 (V110 to L113), and β6 (A114 to H145), each corresponding to a peptide family. Into the same peptide family, it could be supposed that the most concentrated peptides in the hydrolysate would have migrated through the ultrafiltration membrane, in relation with their pI (Appendix A) and the working pH.

For the α1 family, two peptides were recovered (α27–29 and α29–31). Their small sizes could signify that these peptides were final.

The α2 family was represented by 11 recovered peptides. The main fragment was the α33–46, an intermediate peptide, which was hydrolysed in two antimicrobial peptides (α34–46 and α37–46 [14]), and three others with no described activity (α34–43, α37–40, and α41–43), present in the KCl solution. Five other peptides were recovered: α54–59, α58–64, α60–80, α92–98, and α92–97.

The α99–106 family (or α3 family) was essentially composed of the antimicrobial peptide α99–106 [26], which was recovered into the KCl solution with two other peptides derived from α102–106 and α99–105. Another fragment was the α99–107, revealing a different cleavage site of the enzyme contrary to the amino acid in position 106 for the formation of α3 family.

At this low hydrolysis degree, the α107–141 peptide (corresponding to the α4 family) was already digested in other intermediate peptides and its concentration was low in the whole hydrolysate due to the number of generated peptides from its sequence (22). Thus, the preferential produced peptides would migrate. At a working pH of 4.7, the peptide recovery showed 10 well-identified sequences (α107–125, α117–128, α122–127, α129–134, α129–136, α129–141, α129–141, α134–141, α135–141, α136–141, and α137–141), which included two antimicrobial peptides (α129–141 and α137–141). More precisely, the α129–141 and α137–141 peptides were mainly produced from their precursor, the α107–141 peptide [15]. Moreover, the second intermediates from this last peptide were mainly the α107–125 and the α110–128 derived from the α107–128, which was already digested at this hydrolysis degree.

About the β1 family, only one peptide was recovered, the β9–13.

By the same reasoning, the hemorphin family, corresponding to the β2 fragment, was represented mainly by β31–40 and the β32–40 [27], and these peptides were recovered into the KCl solution with β32–36. This family was known for its opioid activity [27].

The small final peptide β82–84 was recovered with β70–79 for the third part of the β-chain.

The β4 family recovery was composed by peptides β92–95 and β102–109, and the derivative of the latter, β103–109.

Finally, the β114–145 fragment, also called the β6 family, was represented mainly by the β114–124 and the β128–145 peptides, as well as their derivatives (β129–145 and β130–139), in this hydrolysate.

Due to the low advanced hydrolysis degree, it was supposed that the majority of the low molecular weight peptides from the precursors were present with a low concentration into the hydrolysate (β35–40 and β34–40 for the β2 family and α99–104 or α101–106 for the α3 family). Thus, it could be supposed that their migration through the ultrafiltration membrane was difficult.

In addition, some peptides with a high pI in the hydrolysate also had a high GRAVY index (i.e., the grand average of hydropathy), indicating their high hydrophobicity. Primarily, it was the case of β103–110 (with a GRAVY index value of 1.5) and β129–137 (with a GRAVY index value of 1.244), which were not recovered in the β4 and β6 families, respectively, in the KCl solution (Figure 4). These peptides could be involved in the hydrophobic interaction with the ultrafiltration membrane or with other hydrophobic peptides [28]. Finally, we can also suppose that an accumulation of cationic peptides on the ultrafiltration membrane surface avoid the migration of the other peptides that were able to migrate [29].

#### 3.1.3. α137-141 Recovery Using pH Control at 4.7

The concentration of α137–141 recovered during EDUF treatment is shown in Figure 2b. pH control allowed the recovery of twice the concentration of α137-141 (from 5.3 ± 0.3 to 10.5 ± 1.5 mg·L^−1^) at the end of the experiment. However, the α137–141 concentration into the recovery compartment did not increase significantly during the two last hours of EDUF treatment under pH control (from 9.3 ± 1.0 to 10.5 ± 1.5 mg·L^−1^). Thereby, a difference between the increase in recovery yield for the total peptides (about 2.8-fold) and for the α137–141 peptide (about twofold) appeared. Figure 2c proposes an explanation. By controlling pH, the α137-141 purity evolution in the recovery compartment showed a maximum after 1 h of the experiment (11.6 ± 2.3%). This purity was not significantly different at 2 h (9.8 ± 0.7%). At the end of EDUF treatment, the recovered α137–141 purity decreased to 6.7 ± 0.9%. In contrast, the α137–141 purity in the recovery compartment increased throughout the experiments without pH control. Then, the maximum α137–141 purity was obtained at 4 h of EDUF treatment (10.3 ± 1.0%). From Figure 2a, we can conclude that the α137–141 migration competed with the continuous migration of other peptides with high molecular weights present in the feed hydrolysate in accordance with its hydrolysis degree. Moreover, it was reported that peptides with high molecular weight can migrate but in a slower way than the low molecular weight peptides because EDUF technology allowed the separation of peptides according to the charge/mass ratio [22]. 

The first part of this work demonstrated that the pH control was an efficient method to improve the peptide yield, with an increase of twofold for the α137–141 peptide recovery.

To improve the α137–141 purity by EDUF, it was necessary to limit the migration of other peptides that were able to co-migrate towards the recovery compartment.

### 3.2. Improvement of the α137-141 Purity in the Recovery Compartment by pH Controlled at 6.5 and 9

After pH of 4.7 was tested, two other pH values were investigated: the first one was a pH close to neutral pH (pH 6.5) and the second was pH 9, close to the pI of the α137–141 peptide (10.5) but lower than pH 10 to avoid basic hydrolysis of feed peptides.

#### 3.2.1. Electrodialytic Parameter Evolution: Conductivity and Current Intensity

Controlling pH at 6.5 or 9 induced a similar tendency for the recovery conductivities (Appendix A) with a first decrease step during the first 2 h with a slope of −0.73 mS·cm^−1^·h^−1^ (*r*^2^ = 0.96) for pH 6.5 and a slope of −0.72 mS·cm^−1^·h^−1^ (*r*^2^ = 0.87) for pH 9. After this, they reached a plateau after 2 h of the experiment. Comparing the pH control at 4.7, the beginning of the second step was delayed from 30 min to 2 h, as previously observed (cf. Section 3.1.1).

As for the current intensity, the evolution tendencies showed the same shape between pH 6.5 and 4.7 (Appendix A), and showed a first decrease step until 2 h, confirming the conductivity analyses.

For the hydrolysate compartment, if the conductivity tendencies were the same, the values were not similar. At pH 6.5, the conductivity value evolved from 10.1 ± 1.5 to 11.6 ± 2.4 mS·cm^−1^, and at pH 9, feed conductivity evolved from 14.6 ± 0.3 to 15.5 ± 0.3 mS·cm^−1^. This difference was due to the NaOH addition after the hydrolysate decolourisation inducing a higher conductivity at a higher working pH (data not shown).

#### 3.2.2. Total Peptide Migration by Controlling pH at 6.5 and 9

According to Figure 5a, the total peptide migration into the recovery compartment decreased significantly as the pH increased. After 4 h of EDUF treatment, the total peptide concentrations were 108.4 ± 2.8 mg·L^−1^ at pH 4.7, 20.8 ± 12.5 mg·L^−1^ at pH 6.5, and 8.2 ± 0.4 mg·L^−1^ at pH 9.

However, except at pH 4.7, the total peptide migration reached a plateau after 2 h of the EDUF experiment. Indeed, the total peptide concentrations during the last 2 h of EDUF treatment evolved from 15.5 ± 6.3 to 20.8 ± 12.5 mg·L^−1^ and from 6.5 ± 0.7 to 8.2 ± 0.4 mg·L^−1^ at pH 6.5 and 9, respectively. All data for the peptide recovery are summarized in Table 1.

To show the results of controlling the pH, Figure 6 compares the chromatograms of the initial feed with those obtained after 4 h of EDUF treatment under the three working pH values. It appeared that a pH increase induced a less effective total peptide migration in terms of both peptide concentration and sequence number.

The mass spectrometry analyses confirmed this striking fact (Appendix A). At the beginning of EDUF treatment, the whole hydrolysate contained 101 identified peptide sequences. Working at pH 4.7 allowed the recovery of 40 peptide sequences, whereas at pH 6.5, this number was divided by two, and only 23 peptide sequences migrated through the ultrafiltration membrane. Last but not least, maintaining the pH at 9 allowed the recovery of only six peptides at significant concentrations (Appendix A).

More precisely, the peptide sequences with a pI lower than the working pH of 6.5 did not migrate into the recovery compartment, except for the α92–98 and α107–125 peptides, which had theoretical pI (6.25 and 6.48, respectively) close to this working pH (6.5). The other recovered peptides were in accordance with the previous condition and they were the same as the peptides obtained by controlling the pH at 4.7. Nevertheless, six peptides that were able to migrate at the working pH of 6.5 were absent in the recovery compartment. These peptides were α58–64; α54–59 for the α2 family; the α102–109, α98–106, and α99–107 from the third fragment of α-chain; and β129–135 from the β4 family (Figure 4). These intermediate peptides were less charged at pH 6.5 than at pH 4.7, and their low concentrations into the hydrolysate were responsible for their absences in the recovery compartment due to the competition with other peptides present in higher concentrations. 

The same explanations may apply for pH 9. In this pH condition, the recovered peptides were in accordance with the recovered peptides using a working pH of 6.5 and with the theoretical peptide pIs, except for β114–124. Nevertheless, β114–124 pI (8.88) was close to the working pH of 9. We obtained β32–40 and β31–40 (from the β2 family), β114–124 (from the β4 family), α99–106 (including the α3 family), and α137–141 and α129–141 (from the α4 family) at significant concentrations in the recovery compartment (Figure 4). For the peptides that were able to migrate but were absent in the recovery compartment, researchers have reported that high pH induces negative charges on ultrafiltration membrane [30] and their charge could increase potential interactions between peptides and membrane [31,32,33]. In this last condition, other peptide sequences were identified in mass spectrometry, but their concentrations were not significant.

#### 3.2.3. α137-141 Migration by Controlling pH at 6.5 and 9

With a pI of about 10.5, the α137–141 peptide was effectively recovered, regardless the values of pH, but its concentration was different according to the pH (Figure 5b). At pH 9, a recovery of 4.9 ± 1.2 mg·L^−1^ after 4 h of EDUF treatment was obtained, whereas concentrations of 6.9 ± 2.4 and 10.5 ± 1.5 mg·L^−1^ were respectively obtained using pH 6.5 and 4.7—the concentration of the α137–141 peptide decreased with an increase in pH. Moreover, after 2 h of EDUF treatment, the α137–141 peptide concentrations reached a plateau and did not evolve significantly until 4 h of the experiment—from 3.4 ± 0.4 to 4.9 ± 1.2 mg·L^−1^ at pH 9, 5.6 ± 1.9 to 6.9 ± 2.4 mg·L^−1^ at pH 6.5, and 9.3 ± 1.04 to 10.5 ± 1.5 mg·L^−1^ at pH 4.7. The decrease in concentration of recovered α137–141 when the working pH increased could have been due to the decrease of peptide charges at higher pH. Secondly, the interaction between the α137–141 peptide, positively charged, and the other peptides, essentially negatively charged, or the α137–141 peptide with membranes, could explain this fact [28,29,31,32,33]. This hypothesis was particularly interesting because when the pH increased, the number of peptides negatively charged in the hydrolysate also increased. Thereby, the probability of interactions between the α137–141 peptide (positively charged) and the other peptides (negatively charged) increased. Consequently, the α137–141 migration through the ultrafiltration membrane would be less important with a higher pH compared with a lower pH. Our results were in accordance with this hypothesis.

Regarding the α137–141 purity, which was one of the main important parameters in this study, we observed in the initial hydrolysate that the α137–141 purity was 0.7 ± 0.1% and its purity increased with an increase in pH during EDUF treatment. At pH 4.7, an enrichment factor of 9 was observed with a final purity of 6.7 ± 0.9%. At pH 6.5, the enrichment factor was about fivefold higher than the obtained purity with the pH controlled at 4.7. Indeed, the enrichment factor was about of 50 corresponding to a α137–141 purity of 37.2 ± 9.9% into the recovery compartment. Nevertheless, the higher purity was obtained for the EDUF treatment at a pH of 9. The α137–141 purity was 56.1 ± 11.3%, which represented a spectacular enrichment factor of 75.

It was important to point out that the α137–141 purity evolution was stopped after 2 h of EDUF treatment for pH values of 6.5 and 9, with respective values of 36.2 ± 7.9% and 50.1 ± 7.9%. This fact was in relation with the observed slowdown of peptide migration after 2 h of the experiment, except for the pH controlled at 4.7.

To conclude this part, we are able to say that maintaining the pH at 9 during the EDUF treatment was an efficient method to obtain the antimicrobial α137–141 peptide with a high purity grade (from 0.7% ± 0.1% in the initial hydrolysate to a maximum of 56.1 ± 11.3% in the recovery compartment). In fact, with 101 initial peptides, this strategy allowed a recovery of α137–141 with only five other peptides in a minor concentration compared to the α137–141 peptide. In accordance with previous papers [8,15,16,17] and the high enrichment factor obtained in α137–141, this first approach is a good way for valorisation of the bovine blood slaughterhouse by-product by application on meat as a food preservative. Accordingly, further studies are in progress to improve the process productivity for the α137–141 and its isolation optimization, in terms of the influence of feed peptide concentration increase or scaling up of membrane area.

## 4. Conclusions

In accordance with the feedstock knowledge, the pH control during EDUF was a performant method to select peptides that migrated into the recovery compartment. 

Increasing hydrolysate pH showed important changes in the peptide transfer though the ultrafiltration membrane. When pH was acidic, more peptides were positively charged and the recovered fraction showed a higher peptide number and a higher total peptide concentration. At the opposite end, when the pH was basic, more peptides were negatively charged, and the recovered fraction showed a lower peptide number and a lower total peptide concentration. Nevertheless, when the pH increased, the EDUF selectivity for the α137–141 recovery increased. Indeed, the maximal enrichment factor compared to the initial hydrolysate was of 75, with a α137–141 purity of about 55% using pH 9. Moreover, only six peptides were recovered when the whole hydrolysate contained more than 100 peptides.

Some commercial antimicrobials are available and efficient at a purity of 2.5% (*w*/*w*) such as nisin. In our case, the purity was very high (more than 50%). Consequently, the obtained fraction could be used with a less quantity for a same effect.

In the current context of food safety, this peptide fraction would be a promising opportunity for the development of efficient, safe, and cost-effective alternative to the synthetic additives used to protect food during its storage and distribution.

## Figures and Tables

**Figure 1 membranes-10-00090-f001:**
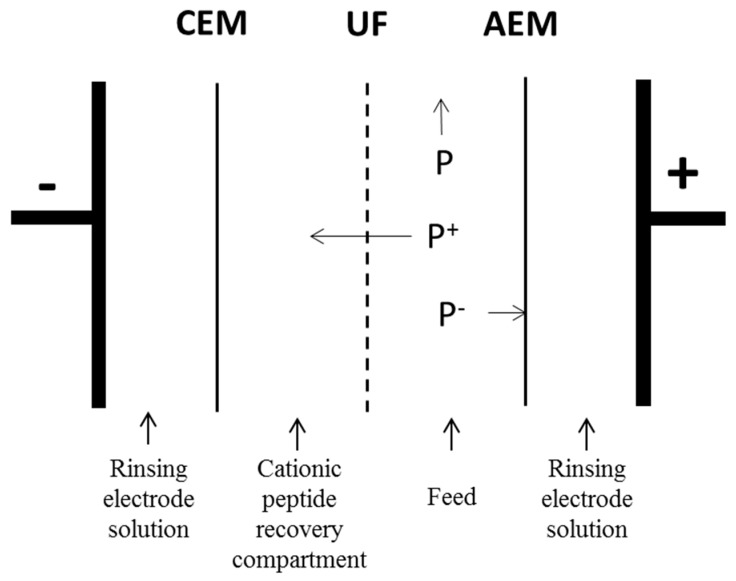
Scheme of electrodialytic cell in cationic configuration. CEM: cation-exchange membrane; AEM: anion exchange membrane; UF: ultrafiltration membrane.

**Figure 2 membranes-10-00090-f002:**
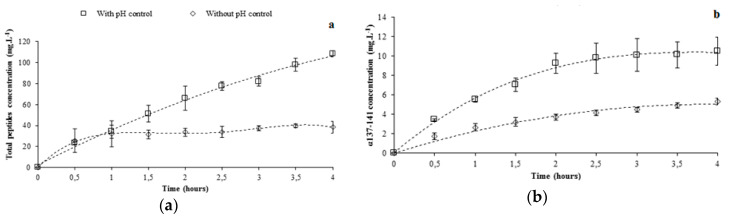
Recovery of total peptides (**a**) and α137-141 (**b**), and evolution of α137-141 rate (**c**) during the electrodialysis with ultrafiltration membrane (EDUF) treatment with and without pH control at 4.7.

**Figure 3 membranes-10-00090-f003:**
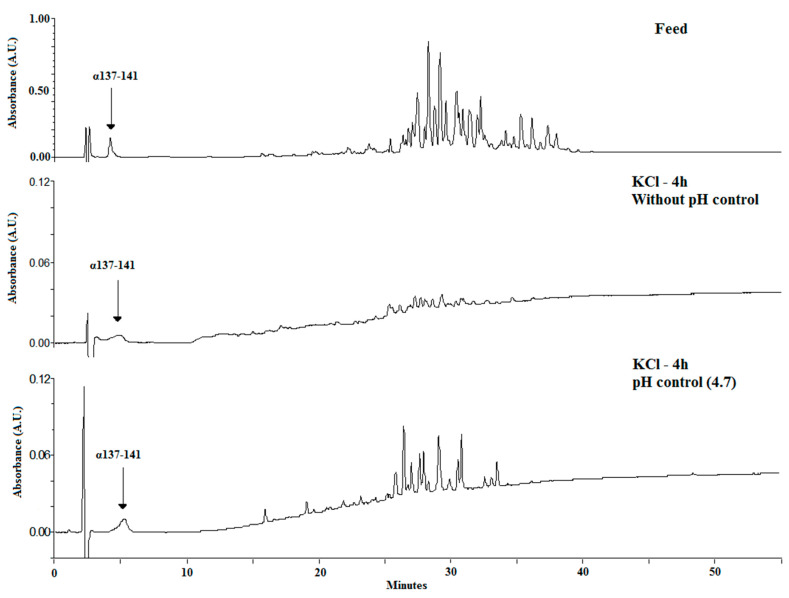
Chromatographic profiles of recovered total peptides after 4 h of EDUF treatment without and with pH control at 4.7.

**Figure 4 membranes-10-00090-f004:**
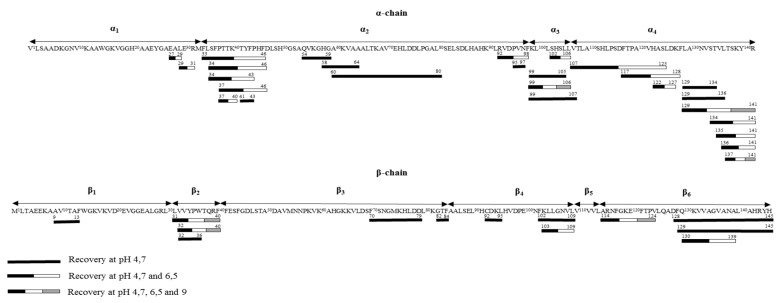
Peptide cartography of pH control recovery (at 4.7, 6.5, and 9) during EDUF treatment.

**Figure 5 membranes-10-00090-f005:**
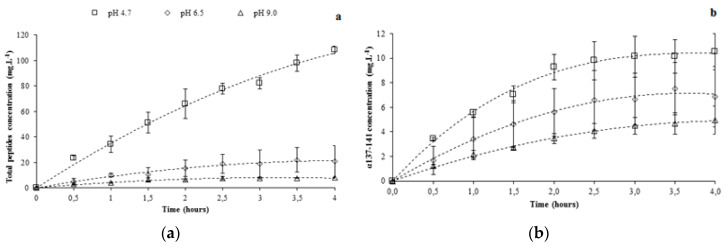
Recovery of total peptides (**a**) and α137-141 (**b**) during the EDUF treatment using pH control (4.7, 6.5, and 9).

**Figure 6 membranes-10-00090-f006:**
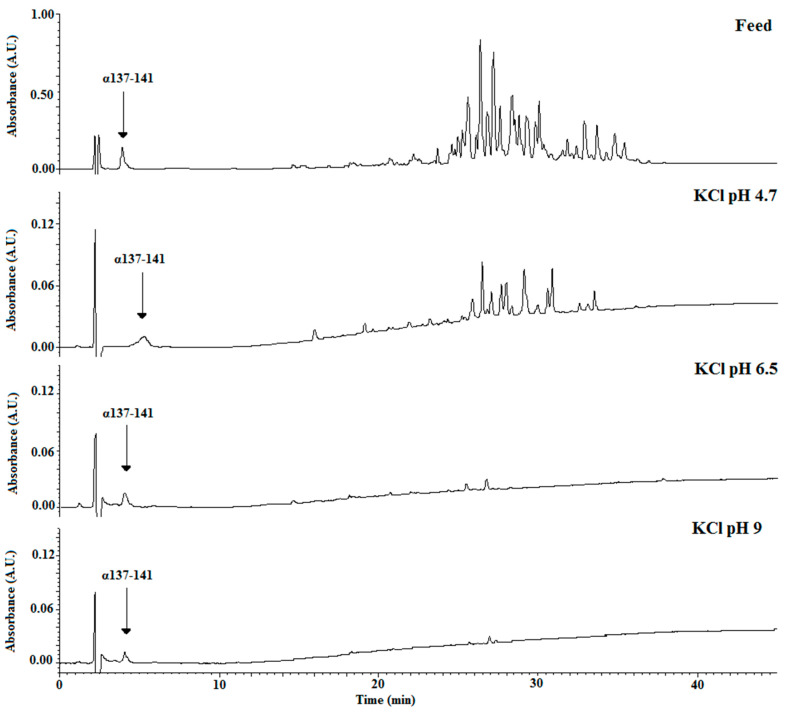
Chromatographic profiles of recovered total peptides after 2 h of EDUF treatment on the basis of the pH control value (4.7, 6.5, and 9).

**Table 1 membranes-10-00090-t001:** Comparison of recovered fractions depending on several working pH values.

pH	No Control	4.7	6.5	9
Total peptide concentration (mg·L^−1^)	38.5 ± 5.5	108.4 ± 2.8	20.8 ± 12.5	8.2 ± 0.4
Number of peptides	16	40	23	6
α137-141 concentration (mg·L^−1^)	5.3 ± 0.3	10.5 ± 1.5	6.9 ± 2.4	4.9 ± 1.2
α137-141 purity (%)	10.3 ± 1.0	6.7 ± 0.9	37.2 ± 9.9	56.1 ± 11.3
α137-141 enrichment factor	-	9	50	75

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
