# Peer review of "Electroseparation of Slaughterhouse By-Product: Antimicrobial Peptide Enrichment by pH Modification"

_membranes, 2020, doi:10.3390/membranes10050090_

Round 1

Reviewer 1 Report

The manuscript includes an approach of the EDUF method for the separation of slaughterhouse by-product, such as bovine blood. The main idea is the preconcentration of the target α137-141 peptide, which is a perspective alternative for synthetic additives to protect food. The authors already have experience in the application of the EDUF to separate molecules in accordance with their charge and molecular weight. This work provides new information about the abilities of the method, such as the recovery of α137-141 peptide.

The Introduction section is clear, and the conclusions are appropriated. I think that this manuscript is publishable in the Journal, and it certainly will have a high impact. The following minor corrections could improve the quality of the manuscript:

  1. The authors provided not much information about the membrane fouling in such solutions. Peptides have a high potential for sedimentation on an IEM surface. Some discussion of the undesirable phenomena should be added.
  2. “The used ultrafiltration membrane (UF) was made of polyethersulfone (PES) with a molecular 105 weight cut-off of 10 kDa (HFK-131, Koch, France).” Why was this membrane chosen? Do its properties correlate with the molecular weight of the targeted peptide? Why wasn’t a triple channel system with two UF membranes with different weight cut-offs used?
  3. I think that there is no need for so many significant digits presented in the article. For example line 178-180 “the total peptide concentrations did not evolve significantly from 2 to 4 hours of the experiment (from 33.75 ± 3.93 to 38.45 ± 5.46 mg.L-1) contrary to the pH control condition (from 66.00 ± 11.70 to 108.36 ± 2.80 mg.L-1)”. The last digit in all numbers does not provide any useful information.
  4. Line 283: the end of the sentence should be as follows …(cf. 3.1.1).

Author Response

Dear reviewer,

Thanks for your evaluation, your comments and your advices.

Please find enclosed our detailed responses.

We hope that our modifications will satisfy you. 

King regards.

Pr Naïma Nedjar-Arroume.

Reviewer 2 Report

This manuscript reported a strategy to increase the purity of one antimicrobial peptide from slaughterhouse by-product hydrolysate by electrodialysis with ultrafiltration membrane. The effect of pH in the separation process was investigated. The idea of this manuscript is interesting, which expand the application field of membrane technology. The manuscript can be accepted after minor revision. Specific comments are as follows: 1. The reason to regulate the pH hydrolysate should be provided in the introduction section. 2. The author should make clear how to conduct pH regulation. Is it online automatic regulation or manual regulation? 3. Does the voltage of electrodialysis affect the separation result? 4. The authors should tell the readers that how much purity is sufficient for commercial use.

Author Response

(The authors gave the same response as above.)

Reviewer 3 Report

This paper provides the analysis of the separation behavior of one specific antimicrobial peptide, α137-141 (653 Da, TSKYR) from bovine blood, a slaughterhouse by-product by electrodialysis with ultrafiltration membrane technique (EDUF) at different pH values.Due to already published papers on the separation of this peptide with the same method, the key of this paper is the importance and effectiveness of the pH control during the EDUF separation as mentioned in Abstract and Introduction.

The work provides valuable new experimental information which confirms the usefulness and validity of EDUF and potential improvement toward higher purity and concentration of the peptide α137-141. For example, it is remarkable that the peptide fraction of the target peptide has increased from 6.69% (at pH 4.7) to 56.13% (at pH 9) after four-hour separation just by controlling the pH of the recovery compartment.

However, the paper lacks a strong rationale for the efficacy of the pH control, and why the values were chosen. There is a need to identify an impact of the other recovered peptides on α137-141. The mechanism for the increase in the performance by pH control is not clear to the reader (what is the physical chemistry behind this step increase?). Furthermore, there is a trade-off between concentration and purity that makes it confusing how pH control can improve the overall efficacy of the process. Also, the conclusion is too short and does not point to the key technical advances of the paper. 

We recommend reconsideration after major revisions. 

Major correction

  • It would be encouraged to have a table of the final concentrations and fraction rates of α137-141 in the recovery compartment at the pH values in order to clearly deliver the main result.
  • While the main focus of this paper is on the analysis of the target peptide, Section 3.1.2 and 3.2.2 deal with total peptide migration, including other peptide components. Since some parts do not seem to have the relevance with the major purpose of the paper, it is recommended to clarify the rationale of these two sections, and their impact on the final conclusions. 

Minor (recommendation)

Another revision on writing check should be carried out to pick out on acronyms and sentence usage. Some examples are:

- The definition and equation for enrichment factor (EF) which is in line of 352, needs to be added.

- In line of 202, DH, the abbreviation form of degree of hydrolysis, was never mentioned before on this paper.

- In line of 334, whatever the values of pH --> in all pH values (for example). "Whatever" should be avoided.

Author Response

Dear reviewer,

Thanks for your evaluation, your comments and your advices.

Please find enclosed our detailed responses.

We hope that our modifications will satisfy you. 

Best regards.

Pr Naïma Nedjar-Arroume.

Round 2

Reviewer 3 Report

The paper has been significantly improved. However, a writing check should be carried out for grammar and English. In particular, there are two sections 5, and in many places the writing/style can be significantly improved. 

Author Response

Author’s reply to the review report #3:

Thanks you for your reviewing and your interesting comments! You can find below our detailed responses. The English of the manuscript has been revised.

This paper provides the analysis of the separation behavior of one specific antimicrobial peptide, α137-141 (653 Da, TSKYR) from bovine blood, a slaughterhouse by-product by electrodialysis with ultrafiltration membrane technique (EDUF) at different pH values.Due to already published papers on the separation of this peptide with the same method, the key of this paper is the importance and effectiveness of the pH control during the EDUF separation as mentioned in Abstract and Introduction.

The work provides valuable new experimental information which confirms the usefulness and validity of EDUF and potential improvement toward higher purity and concentration of the peptide α137-141. For example, it is remarkable that the peptide fraction of the target peptide has increased from 6.69% (at pH 4.7) to 56.13% (at pH 9) after four-hour separation just by controlling the pH of the recovery compartment.

Thanks for your comments. We hope that this work will help other scientists for this kind of challenging separation

However, the paper lacks a strong rationale for the efficacy of the pH control, and why the values were chosen. There is a need to identify an impact of the other recovered peptides on α137-141. The mechanism for the increase in the performance by pH control is not clear to the reader (what is the physical chemistry behind this step increase?). Furthermore, there is a trade-off between concentration and purity that makes it confusing how pH control can improve the overall efficacy of the process. Also, the conclusion is too short and does not point to the key technical advances of the paper.

You are right. The reason for the increase in the performance by pH control is now described in the Introduction (lines 61 to 66):

“Indeed, peptide charges are influenced by the hydrolysate pH. With different studied pH, peptide charges can changed (positively if pH < pI and negatively if pH > pI) and can influenced total migration peptide though the ultrafiltration membrane. In this way, numbers and concentration of peptides being able to migrate in the recovery compartment change with the hydrolysate pH. Consequently, the EDUF treatment selectivity should be influenced by the pH control.”

Moreover, many details about the studied pH were available lines 123 to 129 in Materials and methods section. However, additional precisions have been added for this choice:

“To study the effect of pH on the electroseparation, the compromise was to select different pH that minimize or avoid the migration of peptides which co-migrated with the α137-141 and to keep the possible electrophoretic mobility higher as possible for the α137-141 peptide (pI about 10.5). Moreover, the hydrolysate pH should not exceed 10 to avoid basic hydrolysis. So, the pH was maintained constant at a given value of 4.7 (the native pH-value for the hydrolysate without haem), 6.5 (close to the neutral pH but with an high electrophoretic mobility for α137-141) or 9 by manual addition of HCl (0.1 M) or NaOH (0.1 M) in feed and recovery compartments of EDUF.”

At last, the conclusion has been significantly expanded (lines 481 to 590):

“In accordance with the feedstock knowledges, the pH control in EDUF was a performant method to select peptides that migrated into the recovery compartment.

Increasing hydrolysate showed important changes in the peptide transfer though the ultrafiltration membrane. When pH is acidic, more peptides were positively charged and the recovered fraction showed a higher peptide number and a higher total peptide concentration. At the opposite, when the pH is basic, more peptides were negatively charged and the recovered fraction showed a weaker peptide number and a weaker total peptide concentration. Nevertheless, when the pH increased, the EDUF selectivity for the α137-141 recovery increased. Indeed, the maximal enrichment factor compared to the initial hydrolysate was of 75, with a α137-141 purity of about 55% using pH 9. Moreover, only 6 peptides were recovered when the whole hydrolysate contained more than one hundred peptides.

In the current context of food safety, this peptide fraction would be a promising opportunity for the development of efficient, safe and cost-effective alternative to the synthetic additives used to protect food during its storage and distribution.”

We recommend reconsideration after major revisions.

Thanks for your evaluation. We hope that our modifications in the manuscript will be satisfy you.

Major correction

It would be encouraged to have a table of the final concentrations and fraction rates of α137-141 in the recovery compartment at the pH values in order to clearly deliver the main result.

This is a good idea. A table has been added in the manuscript (table 1, lines 399 to 400). We hope that this table summarizes and clarifies the main results!

Table 1. Comparison of recovered fractions depending on several working pH.

pH

No control

4.7

6.5

9

Total peptide concentration (mg.L-1)

38.5 ± 5.5

108.4 ± 2.8

20.8 ± 12.5

8.2 ± 0.4

Number of peptides

16

40

23

6

α137-141 concentration (mg.L-1)

5.3 ± 0.3

10.5 ± 1.5

6.9 ± 2.4

4.9 ± 1.2

α137-141 purity (%)

10.3 ± 1.0

6.7 ± 0.9

37.2 ± 9.9

56.1 ± 11.3

α137-141 enrichment factor

/

9

50

75

While the main focus of this paper is on the analysis of the target peptide, Section 3.1.2 and 3.2.2 deal with total peptide migration, including other peptide components. Since some parts do not seem to have the relevance with the major purpose of the paper, it is recommended to clarify the rationale of these two sections, and their impact on the final conclusions.

As now available at lines 61 to 66, the study of other peptides is relevant for this work while a particular focus is on the specific migration of the antimicrobial α137-141. Firstly, the total peptide migration showed a different trend than the α137-141 peptide. Secondly, this analysis can explained why, as mentioned lines 450 to 457: “Secondly, the interaction between the α137-141 peptide, positively charged, and the other peptides, essentially negatively charged, or the α137-141 peptide with membranes could explain this fact [28, 29, 31, 32]. This hypothesis was particularly interesting because when the pH increased, the number of peptides negatively charged in the hydrolysate increased too. Thereby, the probability of interactions between the α137-141 peptide (positively charged) and the other peptides (negatively charged) increased. Consequently, the α137-141 migration through the ultrafiltration membrane would be less important with a higher pH compared with a lower pH. Our results were in accordance with this hypothesis.”

At last, the detailed analyse of all peptides is crucial for the final fraction. The characterisation of other peptides helps to guide the potential application of this fraction. Indeed, a powerful antimicrobial as α137-141 could be used if any peptide has a contrary activity (like microbial substrate). Fortunately, this is not the case here.

Minor (recommendation)

Another revision on writing check should be carried out to pick out on acronyms and sentence usage. Some examples are:

- The definition and equation for enrichment factor (EF) which is in line of 352, needs to be added.

Many details have been added in the Materials and methods section (line 141).

“The enrichment factor in α137-141 was assessed by: (Aα137-141/Atot)KCl solution / (Aα137-141/Atot)hydrolysate.”

- In line of 202, DH, the abbreviation form of degree of hydrolysis, was never mentioned before on this paper.

You are right. The abbreviation form of degree of hydrolysis has been removed in Results section to clarify this point (lines 231 to 232):

“Hence, high (intermediate) and low (final) molecular weight peptides were mixed in this hydrolysate at a degree of hydrolysis of 5%.”

- In line of 334, whatever the values of pH --> in all pH values (for example). "Whatever" should be avoided.

In accordance with this point, this sentence has been corrected (lines 442 to 443):

“With a pI of about 10.5, the α137-141 peptide was effectively recovered regarless the values of pH, but its concentration was different according to the pH (figure 5b).”